# Effects of Magnesium Deficiency on Mechanisms of Insulin Resistance in Type 2 Diabetes: Focusing on the Processes of Insulin Secretion and Signaling

**DOI:** 10.3390/ijms20061351

**Published:** 2019-03-18

**Authors:** Krasimir Kostov

**Affiliations:** Department of Pathophysiology, Medical University-Pleven, 1 Kliment Ohridski Str., 5800 Pleven, Bulgaria; dr.krasi_kostov@abv.bg; Tel.: +359-889-257-459

**Keywords:** magnesium deficiency, insulin resistance, type 2 diabetes, insulin secretion, insulin signaling

## Abstract

Magnesium (Mg^2+^) is an essential mineral for human health and plays an important role in the regulation of glucose homeostasis and insulin actions. Despite the widespread clinical evidences for the association of Mg^2+^ deficiency (MgD) and type 2 diabetes mellitus (T2D), molecular mechanisms by which Mg^2+^ contributes to insulin resistance (IR) are still under discussion. Mg^2+^ regulates electrical activity and insulin secretion in pancreatic beta-cells. Intracellular Mg^2+^ concentrations are critical for the phosphorylation of the insulin receptor and other downstream signal kinases of the target cells. Low Mg^2+^ levels result in a defective tyrosine kinase activity, post-receptor impairment in insulin action, altered cellular glucose transport, and decreased cellular glucose utilization, which promotes peripheral IR in T2D. MgD triggers chronic systemic inflammation that also potentiates IR. People with T2D may end up in a vicious circle in which MgD increases IR and IR causes MgD, that requires periodic monitoring of serum Mg^2+^ levels.

## 1. Introduction

Insulin resistance (IR) is associated with an impaired biological response to insulin stimulation of key target tissues, particularly liver, muscle, and adipose tissue. IR impacts glucose utilization, resulting in a compensatory increase in beta-cell insulin production and hyperinsulinemia [1]. Progression of IR can lead to metabolic syndrome (MetS) and type 2 diabetes mellitus (T2D) [2]. According to the International Diabetes Federation, one in every 11 adults has diabetes and T2D accounts for more than 90% of these cases [3]. Globally, 500 million adults are expected to have T2D by 2030 [4].

Magnesium (Mg^2+^) is the fourth most common mineral in the human body, after calcium (Ca^2+^), potassium (K^+^), and sodium (Na+), and the second most abundant intracellular cation after K^+^ [5]. Currently, enzymatic databases list over 600 enzymes for which Mg^2+^ serves as cofactor and an additional 200 in which Mg^2+^ may act as activator [6]. Only 1% of the total Mg^2+^ in the body is present in extracellular fluids and only 0.3% is found in the serum [5]. The normal reference range for Mg^2+^ in the serum is 0.76–1.15 mmol/L. Magnesium deficiency (MgD) is a condition where the serum concentration of Mg^2+^ in the body is ≤0.75 mmol/L (1.8 mg/dL) [6]. Mg^2+^ concentrations ≤0.75 mmol/L may be considered as preclinical hypomagnesemia. Patients are considered frankly hypomagnesemic with serum Mg^2+^ concentrations ≤0.61 mmol/L (1.5 mg/dL). MgD can be present without hypomagnesemia. However, hypomagnesemia, when present, is usually indicative of an important systemic Mg^2+^ deficit [7]. Signs and symptoms of hypomagnesemia usually occur when serum Mg^2+^ is decreased below 0.5 mmol/L (1.2 mg/dL) [7]. A number of factors can negatively affect Mg^2+^ balance in the body and, in the long-term, may result in MgD. Such factors may be a decreased intake of Mg^2+^ from the food or drinking water [8], an increased Mg^2+^ loss through the kidneys [9,10], an impaired intestinal absorption of Mg^2+^ [11], and prolonged use of some medications causing hypomagnesemia [12,13,14].

MgD is associated with an increased risk of multiple preclinical and clinical manifestations, including pancreatic beta-cell dysfunction, IR, increased risk of MetS, and T2D [15,16,17] (Table 1). T2D is often accompanied by alteration of Mg^2+^ status. Intracellular free Mg^2+^ levels are reduced in subjects with T2D, when compared with nondiabetic subjects. An increased prevalence of hypomagnesaemia have been identified in patients with T2D, especially in those with poor glycemic control, with a longer duration of the disease, and with the presence of chronic vascular complications [7]. According to various literature sources, T2D is linked with MgD at an occurrence rate between 13.5–47.7% [18].

## 2. Effects of MgD on Molecular Mechanisms of Insulin Action

Despite the widespread clinical evidence for the association of MgD and T2D, molecular mechanisms by which Mg^2+^ contributes to IR are still under discussion. Currently, the strongest line of evidence supports the effects of MgD on insulin secretion, insulin sensitivity, systemic inflammatory response, and the activity of certain key Mg^2+^-dependent enzymes of carbohydrate and energy metabolism.

### 2.1. Effects of MgD on Insulin Secretion

The insulin producing beta cells are electrically excitable and use changes in membrane potential to couple variations in blood glucose to changes in insulin secretion. After entering the pancreatic beta cells via GLUT2, glucose is converted to glucose-6-phosphate (G6P) by glucokinase (GK). The product of this enzymatic reaction, G6P, is further processed to produce ATP [19]. Subsequent increases in the cytosolic adenosine triphosphate (ATP) / adenosine diphosphate (ADP) ratio control cell membrane potential by inhibiting ATP-sensitive K^+^ (KATP) channels, eliciting a membrane depolarization [20]. An important physiological consequence of the KATP channel closure and the depolarization of the beta-cell membrane is the influx of Ca^2+^ through the L-type Ca^2+^ channels, and insulin release [19] (Figure 1). This occurs when the membrane potential reaches approximately −50 mV [20]. Then, the increase in cytosolic Ca^2+^ is rapidly reversed by very active Ca^2+^ pumps, such as the sarco/endoplasmic reticulum Ca^2+^-ATPase [20]. Secretion of insulin in response to increases in blood glucose levels occurs in two phases [21]. The first phase is rapid and occurs within the few minutes. This phase is followed by a second phase of insulin release, which is more prolonged. Observations show that the first phase of insulin secretion is lost in patients with T2D [22].

The tandem GLUT2 and GK is often referred to as a glucose-sensor controlling blood sugar levels. Mutations that alter the glucose-dependent activity of GK effectively adjust the set point for whole-body glucose homeostasis [20]. Mg^2+^ can directly influence the rate of GK activity by acting as a cofactor for adenine nucleotides (MgATP) [19]. The KATP channel plays a key role in insulin release from pancreatic beta cells. The KATP channel consists of four central inwardly rectifying K^+^ channel (Kir6.2) subunits, surrounded by four regulatory sulfonylurea receptor 1 (SUR1) subunits, whose activity is controlled by the intracellular ATP/ADP ratio [19]. Activating mutations in Kir6.2 or SUR1 subunits of the channel may impair ATP binding at Kir6.2 or may enhance MgADP activation at SUR1. This may disrupt the functioning of KATP channels. Disturbances of this regulation can lead to diabetes in young adults or increase the risk of T2D. These mutations suggest the important role of ATP and MgATP in the regulation of KATP channels [23]. Closure of the KATP channel depends on the binding of ATP to the Kir6.2 subunits. Opening of the KATP channel depends on the binding of MgATP to the SUR1 subunits [19]. The therapeutic target of sulfonylurea drugs (SUs) is the changing of this balance by antagonizing the MgATP binding to the SUR1, which inhibits opening and induces channel closure [23]. In MgD, intracellular levels of ATP and MgATP decrease, which inhibits the closure and opening of KATP channels. This disturbs the coupling between the chemical signal (blood glucose) and the electrical stimulation of the beta-cells, resulting in a disturbance of the normal phases of insulin release. MgD may impair the functioning of the GK, G6P formation, and the accumulation of ATP in beta cells, which may impair closure of the KATP channels. This delays early and late plasma insulin responses to glucose. At physiological extracellular concentrations of Mg^2+^, intracellular levels of MgATP are sufficient for the normal opening of the KATP channels. In MgD, intracellular levels of MgATP decrease, which inhibits the opening of KATP channels. This induces a longer depolarization of the beta-cell plasma membrane followed by the release of more insulin (Figure 1). This effect of extracellular Mg^2+^ on insulin secretion was found in healthy human subjects. In subjects with 0.79 mmol/L plasma Mg^2+^, fasting plasma insulin was 23 μU/mL, while in those with plasma Mg^2+^ 0.87 or 1.00 mmol/L, fasting plasma insulin amounted to 11 μU/mL [24]. From the above, it can be concluded that MgD can disrupt the normal functioning of beta cells and, thus, may trigger beta-cell dysfunction in T2D.

### 2.2. Effects of MgD on Peripheral Insulin Sensitivity

Insulin action begins with the binding of insulin to the insulin receptor (INSR) on the cell membrane of the target cells. The INSR belongs to the family of tyrosine kinase (TK) receptors [25]. INSR is an integral membrane glycoprotein, which is formed by two alpha and two beta subunits. [26]. In the insulin signaling process, insulin binds to the alpha subunit of the receptor that activates the TK in beta subunit. This process also starts autophosphorylation of several tyrosine (Tyr) residues present in the beta subunit. Autophosphorylated residues are then recognized by phosphotyrosine-binding domains of different adaptor proteins belonging to the INSR substrate family (IRS) members (IRS-1 to -6) [27]. The IRS proteins do not contain any intrinsic kinase activity but organize signaling complexes and initiate intracellular signaling pathways [28].

Most insulin actions are carried out by activation of two main signaling pathways, as follows: (1) The Ras/mitogen-activated protein kinases pathway (Ras/MAPK), which regulates gene expression and insulin-associated mitogenic effects, and (2) the phosphatidylinositol-3-kinase (PI3K)/Akt (protein kinase B) pathway, responsible for most its metabolic actions. The PI3K/Akt kinase pathway plays a central role in insulin signaling, since its activation leads to phosphorylation of an important number of substrates with key functions in a wide variety of biological processes, including glucose transport stimulation, glycogen and protein synthesis, and lipogenesis. Akt appears to play an important role in insulin metabolic actions, including muscle and adipose tissue glucose uptake through glucose transporter type 4 (GLUT4) translocation from intracellular compartments to the cell membrane. Additionally, Akt participates in the regulation of glycogen synthesis by glycogen synthase kinase 3 inhibition [26] (Figure 2).

Most common IR alterations include a decrease in the number of INSRs and of their catalytic activity, an increased serine (Ser)/ threonine (Thr) phosphorylation of INSR, an increased Ser/Thr phosphorylation and degradation of IRS proteins, an increase in Tyr phosphatase activity, a decrease in PI3K and Akt kinases activity, and defects in GLUT4 expression and function. These alterations reduce glucose uptake in muscular and adipose tissues and promote alterations at the metabolic level [26,29]. A number of clinical and experimental data indicate that MgD may be associated with a large part of these alterations.

#### 2.2.1. Effects of MgD-Induced Hyperinsulinemia on Downstream Insulin Signaling

In MgD, intracellular levels of Mg^2+^, ATP, and especially of MgATP decrease, as a result of which SUR1 channel subunit cannot be adequately activated. This ultimately leads to an increase in the basal secretion of insulin [19,23] (Figure 3). Basal hyperinsulinemia perpetuates IR by a wide range of mechanisms [30]. These mechanisms are related to the functioning of the INSR and the IRS proteins, which are targets for negative feedback control [31]. Continuous exposure to elevated levels of insulin causes a reduction in the number of receptors exposed on the cell surface by promoting internalization as well as degradation of hormone-occupied receptors. Under conditions of hyperinsulinemia, the receptor’s kinase activity is diminished, probably because of combined effects of phosphorylation of Ser residues on the receptor, dephosphorylation of Tyr residues by the action of phosphatases, and the binding of inhibitory molecules. The IRS proteins also become phosphorylated on Ser and Thr residues, which reduces downstream signaling [30].

#### 2.2.2. Effects of MgD on Activity of the Insulin-Signaling Kinases

Activation of the INSR initiates a cascade of phosphorylation events. The conformational changes and autophosphorylation of the receptor occur at the time of insulin binding, resulting in phosphorylation of receptor substrates such as IRS and Shc proteins. Shc activates the Ras-MAPK pathway, whereas the IRS proteins mostly activate the PI3K-Akt pathway. Activated PI3K leads to the generation of the second messenger phosphatidylinositol-3,4,5-triphosphate (PIP3). Membrane-bound PIP3 activates 3-phosphoinositide dependent protein kinase-1 (PDK1), which phosphorylates and activates Akt and atypical protein kinases. Akt mediates most of insulin’s metabolic effects, regulating glucose transport, lipid synthesis, gluconeogenesis, and glycogen synthesis [32]. Since Mg^2+^ is a necessary cofactor in all ATP transfer reactions, intracellular Mg^2+^ concentrations are critical for the phosphorylation of the INSR and other downstream signal kinases of the target cells [33]. Mg^2+^ participates in a number of reactions related to kinase activation, where it operates together with ATP as a kinase substrate. Additionally, Mg^2+^ has the ability to connect to the regulatory site of the INSR tyrosine kinase (IRTK) and thus can exert regulatory influence. The affinity of IRTK for MgATP increases when the concentration of free Mg^2+^ increases and the IRTK affinity for free Mg^2+^ increases when the MgATP concentration increases [16] (Figure 2). These data suggest that the impairment of kinase activity in MgD may lead to insulin signaling defects and IR (Figure 3).

### 2.3. Effects of MgD on Low-Grade Systemic Inflammation

The inflammatory environment is regarded as an important contributor to IR. Inflammation plays an important role in the development of IR through various cytokines and molecular pathways. Therefore, both the fluctuation of cytokines and the status of relevant signaling pathways should be considered in the analysis of inflammation-associated IR [34]. MgD significantly increases production of various proinflammatory molecules, such as interleukin (IL)-1β (IL-1β), IL-6, tumor necrosis factor-α (TNF-α), vascular cell adhesion molecule-1, and plasminogen activator inhibitor-1, and decreases expression and activity of the antioxidant enzymes such as glutathione peroxidase, superoxide dismutase, and catalase (Figure 3). Cellular and tissue levels of important antioxidants such as glutathione, vitamin C, vitamin E, and selenium are also reduced [35]. IL-1β is a master cytokine that regulates the expression of various other pro-inflammatory cytokines, adipokines, and chemokines. It can induce an inflammatory response and can reduce the expression of IRS-1 at the ERK-dependent transcriptional level and ERK-independent post-transcriptional level. Production of IL-1β is mainly regulated by diet-induced metabolic stress [36]. IL-6 is secreted by multiple tissues, particularly adipose tissue, and is recognized as an inflammatory mediator that causes IR by reducing the expression of IRS-1 and GLUT4. IL-6 also induces IR by blocking the PI3K pathway. Production of IL-6 is regulated by IL-1β [34,36]. Recently, it has been found that serum level of TNF-α is positively correlated with the pathophysiology of IR which exhibits that TNF-α is also a main causative factor that contributes to the development of IR [36]. TNF-α stimulates genes for IL-1α, IL-1β, IL-6, IL-8, IL-18, and cycloxygenase-2 [37]. TNF-α can trigger various transcriptional pathways such as nuclear factor kappa B (NF-κB) and c-Jun NH2-terminal kinases (JNK). Once NF-κB and JNK are activated, they phosphorylate Ser307 in IRS-1 and this may result in impairment of INSR-mediated Tyr phosphorylation of IRS-1 [36]. In addition, TNF-α may reduce protein levels of GLUT4, along with a decreased activity of Akt in adipocytes [38]. Chemokines are an important class of pro-inflammatory mediators. Their production is dependent on the activation IL-1β and various transcriptional pathways. Оverexpression of monocyte chemotactic protein-1 (MCP-1) in adipose tissues was observed to be responsible for the increase in adipose tissue macrophages and induction of IR [36].

The results of several clinical studies have shown that increased synthesis and release of proinflammatory cytokines may be the link between obesity and MetS. On the other hand, hypomagnesemia triggers low-grade chronic inflammation and Mg^2+^ deficit may be associated with the development of MetS [39]. Guerrero-Romero et al. found a link between Mg^2+^ levels, inflammation, and oxidative stress, as risk factors for the development of MetS [15]. These findings support the hypothesis that MgD can play an important role in the pathophysiology of MetS and the actuation of the inflammatory reaction caused by the shortage of Mg^2+^ could be the link between MgD and MetS [39]. There is a strong correlation between mass of adipose tissue and development of IR in T2D. Corica et al. have recently shown that patients with T2D, having abdominal obesity, a lipid profile with high risk, and high blood pressure, have lower levels of Mg^2+^, compared with patients without metabolic risk factors. Furthermore, plasma triglycerides and waist circumference were independently associated with hypomagnesaemia [15]. An important role in obesity is the endocrinological function of adipocytes. Various effectors such as leptin, adiponectin, IL-1, IL-6, IL-8, IL-18, TNF-α, resistin, ghrelin, visfatin, orexin, adipsin, and cortisol are produced by adipocytes or are related to obesity [16]. The most important adipose derived mediators that cause IR are free fatty acids (FFAs) and adipokines. Recent evidence has suggested that saturated FFAs may act through binding to toll-like receptor 4. This binding leads to the activation of NF-κB and activator protein-1 transcription factors and these lead to an increased production of proinflammatory cytokines, such as TNF-α, IL-6, and IL-1 in adipose tissue [38]. Adipokines are a large group of pro-inflammatory mediators that include leptin, TNF-α, IL-6, and MCP-1. Adiponectin is the only adipokine that has anti-inflammatory action. The level of adiponectin is downregulated in obesity and correlates positively with insulin sensitivity. The imbalance between leptin and adiponectin may be an important cause for the development of IR [36].

MgD is often found in people with MetS and T2D, which are connected with higher plasma concentrations of C-reactive protein (CRP) [40]. Low Mg^2+^ intake is associated with a higher probability of increased serum CRP levels in children [41]. There is also an association between the dietary intake of Mg^2+^ and elevated CRP levels in the adult population [42]. Intake of Mg^2+^ is also inversely related to the level of high-sensitivity CRP, IL-6, and fibrinogen [43]. CRP is an acute-phase protein synthesized by the liver. It is an inflammatory marker whose expression is increased significantly during inflammation, mainly due to its regulation by proinflammatory cytokines such as IL-6 and TNF-α. Elevated CRP levels may be a potential risk factor or marker for vascular complications in T2D. However, there is no clear causality between serum CRP, IR, and T2D, which suggests that CRP is more likely to be a downstream marker that links inflammation to IR [34].

Another inflammatory mediator that has been recently implicated in the pathogenesis of several chronic diseases, including MetS and T2D, is the lipid mediator leukotriene B4 (LTB4). LTB4 promotes IR in adipose tissue directly and by enhances the production of other proinflammatory cytokines [44].

### 2.4. Effects of MgD on Key Mg^2+^-Dependent Enzymes of Carbohydrate and Energy Metabolism

MgD can also be a rate-limiting factor in carbohydrate and energy metabolism, since many of the enzymes in these processes require Mg^2+^ or MgATP as a cofactor during reactions that use the phosphorus bond (Figure 3) [45]. In pancreatic beta cells Mg^2+^ can directly influence GK activity because the enzyme’s action depends on MgATP [19]. In the liver, Mg^2+^ is an important regulator of enzymes in gluconeogenesis. Phosphoenolpyruvate carboxy kinase (PEPCK), fructose-1,6-bisphosphatase (F1,6BP), pyruvate carboxylase (PC), and glucose-6-phosphatase are rate limiting enzymes in this process. Of these four enzymes, Mg^2+^ is required by three, those are PC, PEPCK, and F1,6BP [46]. Glycogen synthase kinase 3 (GSK3) is a key enzyme that regulates glycogen synthase (GS). Insulin signaling was the first pathway found to increase the inhibitory serine phosphorylation of GSK3 by activation of PI3K and Akt [47]. GSK3 is a constitutively active enzyme and hormones and substances that mediate signaling through GSK3 pathways typically trigger a reduction in kinase activity. The insulin signal transduction pathway provides a clear example of this. GSK3 phosphorylates GS at a cluster of residues (Ser641, Ser645, and Ser649) to maintain it in a dormant state. Insulin causes GSK3 to become phosphorylated at Ser21 of GSK3α and at the homologous position, Ser9, in GSK3β. These are inhibitory post-translational modifications that diminish the kinase activity of GSK3, so insulin signaling relieves GSK3-mediated suppression of GS activity and results in the synthesis of glycogen [48]. Mg^2+^ divalent cations are ligands with GSK3 functioning to balance the negative charge that is on the aspartic acid side chains. Asp200 and Asp264 both have side chains with net negative charges and Mg^2+^ cations can neutralize their negative charge. This process can be inhibited by lithium (Li^+^) because its cations directly compete with the Mg^2+^ cations. Li^+^ is a mood stabilizer, widely used in the chronic treatment of bipolar disorders. Li^+^ ions directly inhibit GSK-3 by interrupting the ligand complex with Mg^2+^ [49]. Mg^2+^ is an important controller of glycolysis and the Krebs cycle. Mg^2+^ regulates important glycolytic enzymes, such as hexokinase, phosphofructokinase, phosphoglycerate kinase, and pyruvate kinase. The most important effect is attributable to the MgATP complex, which is a cofactor for these enzymes [50,51]. Mg^2+^ has been documented to enhance the activity of three important mitochondrial dehydrogenases involved in energy metabolism. The activities of isocitrate dehydrogenase and 2-oxoglutarate dehydrogenase complex (OGDH) are stimulated directly by the Mg^2+^-isocitrate complex and free Mg^2+^, respectively. The activity of pyruvate dehydrogenase complex is stimulated indirectly via the stimulatory effect of Mg^2+^ on pyruvate dehydrogenase phosphatase, which dephosphorylates and, thus, activates the pyruvate decarboxylase. The results indicate that OGDH is the main step of oxidative phosphorylation modulated by Mg^2+^ when 2-oxoglutarate is the oxidisable substrate. Mg^2+^ is also an activator of ATP synthesis by mitochondrial F0/F1-ATPase [50].

## 3. Genetic Relationships between MgD and T2D

Intracellular Mg^2+^ concentrations are determined by various Mg^2+^ channels and transporters. Of these, transient receptor potential melastatin type 6 and 7 (TRPM6 and TRPM7) ion channels, solute carrier family 41 member 1 (SLC41A1), and Mg^2+^ transporter 1 (MagT1) play a major role. Several groups of researchers have investigated the association between genetic variations in these Mg^2+^ transporters and risk for T2D. Until now, single nucleotide polymorphisms have been found in the TRPM6 and SLC41A1 genes, which are associated with an increased risk for T2D [19]. Patients with dominant mutations in the hepatocyte nuclear factor 1B (HNF1B) gene or recessive mutations in the pterin-4 alpha-carbinolamine dehydratase 1 (PCBD1) gene can develop hypomagnesemia and maturity-onset diabetes of the young [9,52,53]. Also, the genetic changes of the pancreatic beta-cell KATP channels can play a central role in the change of insulin secretion. Common variants in the potassium inwardly rectifying channel, subfamily J, member 11 (KCNJ11) and ATP-binding cassette, sub-family C, member 8 (ABCC8) genes, that encode channel subunits Kir6.2 and SUR1, were identified as T2D susceptibility loci. KATP channel activity is regulated by the intracellular balance of MgATP and MgADP and depends on the presence of Mg^2+^ ions. Low Mg^2+^ intake and imbalance in intracellular Mg^2+^ concentrations may interact with KATP ion channel variants in affecting T2D risk [54] (Figure 3).

## 4. Main Causes and Risk Factors for MgD

A number of factors can negatively affect Mg^2+^ balance in the body and, in the long-term, may result in MgD. Such factors may be a decreased intake of Mg^2+^ from the food or drinking water, an increased Mg^2+^ loss through the kidneys, an impaired intestinal absorption of Mg^2+^, and the use of some medications (Figure 3).

### 4.1. Decreased Intake of Mg^2+^ from the Food or Drinking Water

Recommended Dietary Allowance (RDA) for Mg^2+^ is 420 mg/day for men and 320 mg/day for women [55]. In the Western world, dietary intake of Mg^2+^ is subnormal, with shortfalls of between 65 and 225 mg of Mg^2+^/day, depending upon geographic region [56]. Dietary studies in Europe and the United States of America (USA) show that people who consume Western-style diets have low Mg^2+^ content, <30–50% of the RDA for Mg^2+^. It is assumed that the Mg^2+^ intake in the USA has decreased over the past 100 years, from about 500 mg/day to 175–225 mg/day [57]. Dietary intake of Mg^2+^ has also been shown to be insufficient in the general United Kingdom population [58] and in middle-aged French adults [59]. This is probably the result of the increasing use of processed foods [57]. Refining or processing of food may deplete Mg^2+^ content by nearly 85%. Furthermore, the incidence rate of MgD can vary considerably in different regions due to the large differences of Mg^2+^ content in drinking water, which can provide up to 30% of daily needs [60]. Water is a variable source of Mg^2+^ intake. Since this varies depending on the area from which water comes, Mg^2+^ intake from water is usually not estimated to a sufficient extent [61]. In drinking water, the levels of Mg^2+^ should be at least 10 mg/L and ideally 25–100 mg/L [62]. Therefore, it seems reasonable to assume that MgD is mainly related to the low intake of Mg^2+^ in food and drinking water, which may lead to a negative balance over time [60]. Inadequate dietary intake of Mg^2+^ is an independent risk factor for the development of T2D. Lopez-Ridaura et al., evaluating 37,309 participants free of cardiovascular disease and T2D, found a significant inverse association between Mg^2+^ intake and diabetes risk [63]. Van Dam et al. reported a similar relationship. Their findings indicated that a diet high in Mg^2+^-rich foods, particularly whole grains, is associated with a substantially lower risk of T2D [64].

### 4.2. Increased Loss of Mg^2+^ through the Kidneys

Mg^2+^ homeostasis is governed by reabsorption of Mg^2+^ from primary urine in the kidney. Of about 2,400 mg Mg^2+^ that is ultra-filtrated daily, 95–99% is reabsorbed by the nephrons. The causes of Mg^2+^ loss through the kidneys are elucidated in detail during the recent years and may involve various damage of the transport systems located in the thick ascending limb (TAL) of Henle’s loop and distal convoluted tubules [6,12]. Inherited tubular disorders that result in urinary Mg^2+^ waste are Gitelman syndrome, Bartter syndrome, familial hypomagnesemia with hypercalciuria and nephrocalcinosis (FHHNC), autosomal-dominant hypocalcemia with hypercalciuria (ADHH), isolated dominant hypomagnesemia (IDH) with hypocalciuria, isolated recessive hypomagnesemia (IRH) with normocalcemia, and hypomagnesemia with secondary hypocalcemia (HSH) [9,65,66]. Recent findings in obese diabetic rats found that TRPM6 was down regulated, explaining renal Mg^2+^ waste. Mg^2+^ uptake in the TAL of Henle may also have a role, since hyperinsulinemia and IR may lead to a decreased Mg^2+^ uptake and an increase Mg^2+^ excretion. Hyperglycemia also leads to increased urinary Mg^2+^ waste, contributing to Mg^2+^ depletion. Plasma Mg^2+^ levels were found inversely correlated with the urinary Mg^2+^ excretion rate and with fasting blood glucose values, suggesting that the tubular reabsorption of Mg^2+^ is decreased in the presence of severe hyperglycemia [67]. Several drugs, such as loop diuretics (including furosemide, bumetanide, and ethacrynic acid), produce large increases in Mg^2+^ excretion through the inhibition of the electrical gradient necessary for Mg^2+^ reabsorption in the TAL. Long-term thiazide diuretic therapy also may cause MgD, through enhanced Mg^2+^ excretion and, specifically, reduced renal expression levels of the epithelial Mg^2+^ channel TRPM6. Many nephrotoxic drugs, including cisplatin, cyclosporine, and tacrolimus, can produce urinary Mg^2+^ waste by a variety of mechanisms, some of which are still unknown [12].

### 4.3. Impaired Intestinal Absorption of Mg^2+^

Intestinal Mg^2+^ absorption occurs predominantly in the small intestine via a paracellular pathway and smaller amounts are absorbed into the colon, mainly via a transcellular pathway. Paracellular Mg^2+^ absorption occurs through simple diffusion and involves the transport of Mg^2+^ through small spaces between epithelial cells and depends on specialized transmembrane proteins, claudins, which control ion permeability. The transcellular pathway of Mg^2+^ absorption depends on the TRPM6 and TRPM7 transporters. Various factors influence the intestinal uptake of Mg^2+^ and are of substantial importance for the supply of this mineral [68]. Impaired gastrointestinal Mg^2+^ absorption is a common underlying basis for hypomagnesemia, especially when the small bowel is involved, due to disorders associated with malabsorption, chronic diarrhea, or steatorrhea, or as a result of bypass surgery on the small intestine. As there is some Mg^2+^ absorption in the colon, patients with ileostomies can develop hypomagnesemia [69]. Mg^2+^ absorption may be impaired in a number of diseases, e.g., Crohn’s disease, ulcerative colitis, coeliac disease, short bowel syndrome, and Whipple’s disease [11,57]. Genetic modification of the TRPM6 channel was identified as the underlying cause for the autosomal recessive disorder HSH. Individuals with HSH fail to effectively absorb Mg^2+^ when the intraluminal intestinal concentration of Mg^2+^ is low [70]. Proton pump inhibitors (PPIs) are commonly used medicines, to reduce gastric acid secretion, that can affect the gastrointestinal absorption of Mg^2+^. Hypomagnesemia is a recognized side-effect of PPIs [13]. Use of metformin to treat T2D may also cause hypomagnesaemia due to gastrointestinal losses, such as the risk increases with the duration of the therapy [14]. This may be an important cause of the higher incidence of MgD among the T2D population.

## 5. Mg^2+^ Supplementation and Dietary Approaches for Improving Insulin Sensitivity in T2D

Benefits of Mg^2+^ supplementation in diabetic subjects have been found in a number of clinical studies. They demonstrate that oral administration of Mg^2+^ reduces IR and improves insulin sensitivity in T2D patients, as well as in overweight nondiabetic subjects (Table 2).

These data indicate that special attention should be given to the risk groups, in particular individuals with MetS and T2D, in which serum Mg^2+^ levels should be monitored periodically. The current RDA for Mg^2+^ ranges from 80 mg/day for children 1–3 years of age to 130 mg/day for children 4–8 years of age. For older males, the RDA for Mg^2+^ ranges from as low as 240 mg/day (range, 9–13 years of age) and increases to 420 mg/day for males 31–70 years of age and older. For females, the RDA ranges from 240 mg/day (9–13 years of age) to 360 mg/day for females 14–18 years of age. The RDA for females 31–70 years of age and older is 320 mg/day. Many nutritional experts feel the ideal intake for Mg^2+^ should be based on the body weight (e.g., 4–6 mg per kg/day) [57]. Organic bound Mg^2+^ salts, such as Mg^2+^ citrate, gluconate, orotate, or aspartate, are recommended in the treatment of MgD because of their high bioavailability [57,78]. A potential side effect from the use of Mg^2+^ salts may be the risk of developing hypoglycemia by increasing the intestinal absorption rate of SUs. Therefore, SUs should be given at least 1 h before Mg^2+^ intake [79].

Considering the numerous positive effects of Mg^2+^ on a number of mechanisms related to IR, consuming a healthy Mg^2+^-rich diet should be encouraged for individuals with MetS and T2D. There is consensus on the benefits of certain named dietary patterns, such as the Mediterranean diet and DASH (Dietary Approaches to Stop Hypertension) diet for prevention and management of T2D [80]. The Mediterranean diet is rich in Mg^2+^, dietary fiber, antioxidants, and polyphenolic compounds [81]. The DASH eating plan is an acceptable dietary pattern for people who have diabetes. In addition to promoting blood pressure control, this eating pattern has been shown to improve IR, hyperlipidemia, and even overweight/obesity [82]. The DASH diet contains larger amounts of Mg^2+^, K^+^, Ca^2+^, dietary fiber, and protein and smaller amounts of total and saturated fat and cholesterol than the typical diet [83].

## 6. Conclusions

Maintaining serum Mg^2+^ concentrations within the reference range is essential for normal insulin secretion and activity, as well as for the optimal functioning of many enzymes of glucose and energy metabolism. MgD may be associated with beta-cell dysfunction, IR, reduced glucose tolerance, and ultimately, clinical manifestations of T2D. Oral Mg^2+^ supplementation and appropriate dietary patterns improve insulin sensitivity and metabolic control in patients with T2D, suggesting that Mg^2+^ is an important factor in the etiology and management of this widespread socially significant disease.

## Figures and Tables

**Figure 1 ijms-20-01351-f001:**
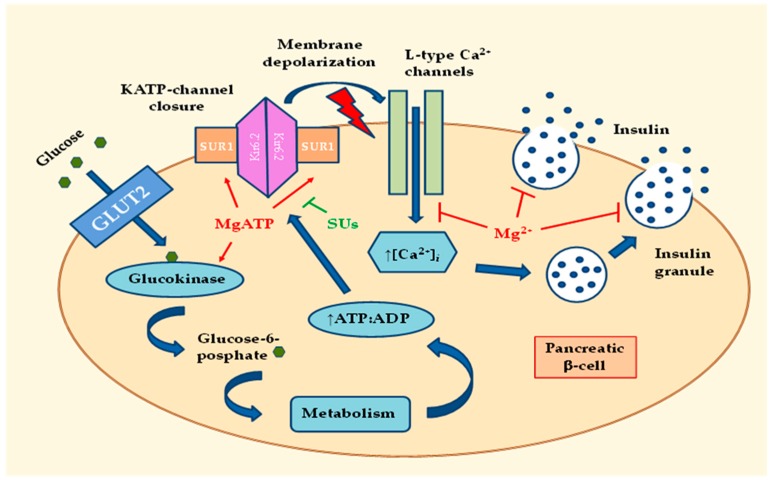
Regulatory role of Mg^2+^ in the insulin secretion from pancreatic beta cells. The normal intracellular Mg^2+^ concentrations are of utmost importance for the optimal insulin secretion. The first step of beta-cell glucose metabolism is the conversion of glucose to glucose-6-phosphate by glucokinase, which subsequently results in a rise in intracellular ATP. MgD can directly influence the rate of GK activity because the enzyme’s action depends on MgATP. Closure of the KATP channel depends on the binding of ATP to the Kir6.2 subunits. Opening of the KATP channel depends on the binding of MgATP to the SUR1 subunits. An important consequence of the closure of KATP channels is the depolarization of the beta-cell membrane, which stimulates Ca^2+^ influx through L-type Ca^2+^ channels and insulin release. In MgD, intracellular levels of ATP and MgATP decrease. This disturbs the coupling between the chemical signal (blood glucose) and the electrical stimulation of the beta cells, resulting in a disturbance of the normal phases of insulin release. SUs antagonize the binding of MgATP to the SUR1, which induces channel closure and potentiates insulin secretion. Legend: GLUT2, glucose transporter type 2; KATP, ATP-sensitive K^+^ channel; SUR1, sulfonylurea receptor 1 subunit of KATP; Kir6.2, inwardly rectifying K^+^ channel subunit of KATP; SUs, sulfonylurea drugs; Mg^2+^, magnesium; MgATP, Mg^2+^-ATP complex; ↑[Ca^2+^]*i*, increased intracellular Ca^2+^ concentrations; ↑ATP:ADP, increased ATP/ADP ratio.

**Figure 2 ijms-20-01351-f002:**
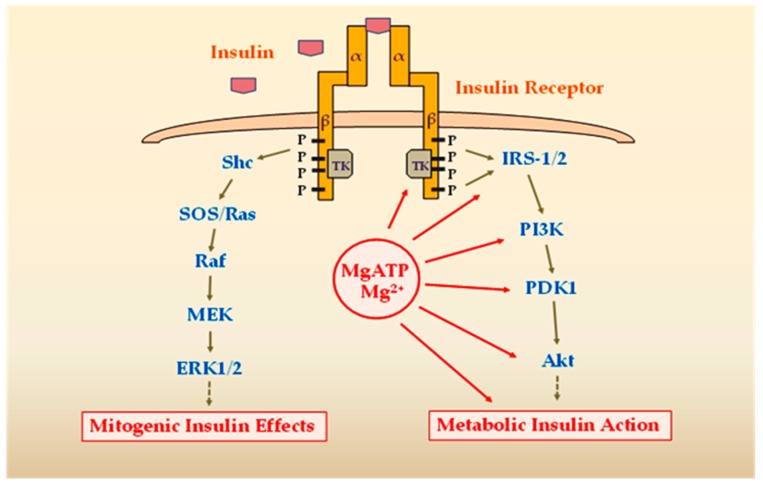
Schematic diagram showing the two major insulin signaling pathways and the role of Mg^2+^ in metabolic signaling. The Ras/MAPK pathway regulates gene expression and insulin-associated mitogenic effects. The PI3K/Akt kinase pathway regulates the metabolic actions of insulin, which include glucose uptake by GLUT4 mobilization, glycogen and protein synthesis, and lipogenesis. Intracellular Mg^2+^ concentrations are critical for the phosphorylation of INSR and the activity of other signal kinases, where Mg^2+^ operates together with ATP as a kinase substrate. Mg^2+^ may exert regulatory influence on TK of INSR and other enzymes, mediating the metabolic effects of insulin. Legend: TK, tyrosine kinase; IRS, insulin receptor substrate; PI3K, phosphatidylinositol-4,5-bisphosphate-3-kinase; PDK1, 3-phosphoinositide dependent protein kinase-1; Akt, protein kinase B; Shc, SH2 domain-containing protein; SOS, son of sevenless; MEK, mitogen-activated extracellular signal-regulated kinase; ERK1/2, extracellular signal-regulated kinase 1/2; MgATP, Mg^2+^-ATP complex; Mg^2+^, magnesium.

**Figure 3 ijms-20-01351-f003:**
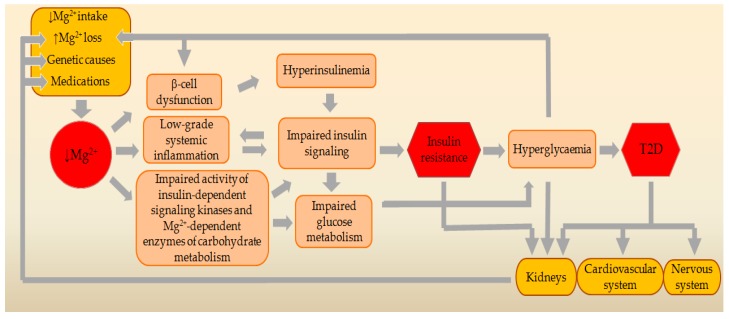
Schematic presentation of the pathogenetic vicious circle between MgD and IR in T2D. A number of factors can negatively affect Mg^2+^ balance in the body and may result in MgD. MgD may cause beta-cell dysfunction, impaired insulin signaling, development of chronic systemic inflammation, and change in activity of key Mg^2+^-dependent signaling kinases and enzymes of carbohydrate and energy metabolism. This can ultimately contribute to development of IR and T2D. Legend: ↓Mg^2+^, magnesium deficiency; T2D, type 2 diabetes mellitus.

**Table 1 ijms-20-01351-t001:** Clinical manifestations of MgD.

**General:** Anxiety, agitation, irritability, headache, loss of appetite, and nausea.
**Musculature:** Muscle spasm and tetany.
**CNS/Nerves:** Nervousness, migraine, depression, poor memory, low stress tolerance, paraesthesia, tremor, and seizures.
**Metabolism:** Pancreatic beta-cell dysfunction, IR, decreased glucose tolerance, increased risk of MetS and T2D, dyslipoproteinemia, disorders of vitamin D metabolism, resistance to PTH, and osteoporosis.
**Cardiovascular system:** Arrhythmias, coronary spasm, atherosclerosis, hypertension, arterial stiffness, endothelial dysfunction, and increased platelet aggregation.
**Electrolytes:** Sodium retention, hypokalemia, and hypocalcemia.

**Table 2 ijms-20-01351-t002:** The benefits of Mg^2+^ supplementation for improvement of insulin sensitivity in patients with T2D and overweight non-diabetic subjects.

Study	Mg^2+^ Intake (mg/day)	Results
Rodrguez-Moran et al., 2003 [71]	50 mL MgCl_2_ solution (50 g MgCl_2_ per 1000 mL solution) daily for 16 weeks.	Oral supplementation with MgCl_2_ solution restores serum Mg^2+^ levels, improving insulin sensitivity and metabolic control in T2D patients with decreased serum Mg^2+^ levels
Guerrero-Romero et al., 2004 [72]	MgCl_2_ 2.5 g daily for 3 months	Oral Mg^2+^ supplementation improves insulin sensitivity in hypomagnesemic non-diabetic subjects
Song et al., 2006 [73]	360 mg/day for 4–16 weeks	Oral Mg^2+^ supplementation reduces plasma fasting glucose levels and increases HDL cholesterol in patients with T2D
Chacko et al., 2011 [74]	500 mg/day for 4 weeks	Mg^2+^ treatment significantly improves fasting C-peptide concentrations and fasting insulin concentrations
Mooren et al., 2011 [75]	365 mg/day for 6 months	Mg^2+^ supplementation resulted in a significant improvement in fasting plasma glucose and insulin sensitivity in normomagnesemic, overweight non-diabetic subjects
Solati et al., 2014 [76]	300 mg/day for 3 months	Oral Mg^2+^ supplementation has beneficial effects on blood glucose, lipid profile, and blood pressure in patients with T2D
ELDerawi et al., 2019 [77]	250 mg/day for 3 months	Oral Mg^2+^ supplementation reduces IR and improves glycemic control in T2D patients

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
