# Peer review of "Effects of Magnesium Deficiency on Mechanisms of Insulin Resistance in Type 2 Diabetes: Focusing on the Processes of Insulin Secretion and Signaling"

_ijms, 2019, doi:10.3390/ijms20061351_

Round 1
Reviewer 1 Report
The author has collected information from previously published articles on the role of Magnesium in regulation of glucose homeostasis and insulin actions. Current article has interesting collection of work with several in-vitro, in-vivo and clinical studies which, is beneficial to researchers in the areas of Insulin resistance and diabetes research. However, the current form of article needs some revisions, and the author is advised to edit the manuscript professionally before submission.
Some inadequacies and suggestions listed below:
It is advised to reframe some sentences to avoid many and/commas ( , & and)
1.Abstract Line 12-13: Mg2+ regulates glucokinase activity, ATP-sensitive channels and L-type Ca2+ channels in pancreatic beta cells, preceding insulin secretion.
Line 13-14: Intracellular Mg2+ concentrations are critical for the phosphorylation of all ATP, phosphate transfer-associated enzymes and tyrosine kinases including insulin receptor tyrosine kinase.
2. It is suggested to avoid the repeated information in the article (ex: line 58-61 and 117-119, about fate of glucose)
3. Line 55- 102: information presented is well known and the author did not focus on role of Mg2+ on insulin secretion. However, it is covered in 3.1 and subheading 2 does not have much importance in this article.
4. Figure legend, Line 104, …schematic diagram of the …
5. In several places of the article, sentences lack the connection from one to another.
6. It is suggested to include cartoons/illustrations on how does Mg2+ regulates insulin secretion in pancreatic β-cells and how does Mg2+ affects insulin sensitivity?
7. Elaborate the Figure legend 2 in few lines
Author Response
Response to Reviewer 1
It is advised to reframe some sentences to avoid many and/commas ( , & and)
1. Abstract Line 12-13: Mg2+ regulates glucokinase activity, ATP-sensitive channels and L-type Ca2+ channels in pancreatic beta cells, preceding insulin secretion.
Line 13-14: Intracellular Mg2+ concentrations are critical for the phosphorylation of all ATP, phosphate transfer-associated enzymes and tyrosine kinases including insulin receptor tyrosine kinase. (line 13-15)
2. It is suggested to avoid the repeated information in the article (ex: line 58-61 and 117-119, about fate of glucose) (Subheading 2 has been removed, which eliminates the repeated information in the article.)
3. Line 55- 102: information presented is well known and the author did not focus on role of Mg2+ on insulin secretion. However, it is covered in 3.1 and subheading 2 does not have much importance in this article. (Subheading 2 has been removed)
4. Figure legend, Line 104, (now is figure 2) …schematic diagram of the … (line 148-159)
5. In several places of the article, sentences lack the connection from one to another. (?)
6. It is suggested to include cartoons/illustrations on how does Mg2+ regulates insulin secretion in pancreatic β-cells (line 107-124) and how does Mg2+ affects insulin sensitivity? (line 146-159)
7. Elaborate the Figure legend 2 (now is figure 3) in few lines (line 299-304)
Reviewer 2 Report
This is a review article regarding Mg deficiency and MetS or T2DM. The manuscript was concisely and well-written. There are a few comments.
1. In addition to IR, the title may include insulin secretion or beta cell function as well.
2. Treatment/intervention of MgD is important. Results of clinical studies examining efficacy of Mg supplementation should be described more precisely and in detail. The results may be better to summarize in a table.
Author Response
1. In addition to IR, the title may include insulin secretion or beta cell function as well. (line 2-4)
2. Treatment/intervention of MgD is important. Results of clinical studies examining efficacy of Mg supplementation should be described more precisely and in detail. The results may be better to summarize in a table. (line 396-398)
Round 2
Reviewer 1 Report
Author has addressed all points mentioned in the previous version of the article.
The current form of review is well presented, and I have no additional comments.
Reviewer 2 Report
The manuscript has been revised properly.